# Inedible Food Waste Linked to Diet Quality and Food Spending in the Seattle Obesity Study SOS III

**DOI:** 10.3390/nu13020479

**Published:** 2021-01-31

**Authors:** Shilpi Gupta, Chelsea M. Rose, James Buszkiewicz, Jennifer Otten, Marie L. Spiker, Adam Drewnowski

**Affiliations:** 1Center for Public Health Nutrition, University of Washington, Seattle, WA 98105, USA; cmr013@uw.edu (C.M.R.); buszkiew@uw.edu (J.B.); jotten@uw.edu (J.O.); mspiker@uw.edu (M.L.S.); adamdrew@uw.edu (A.D.); 2Department of Epidemiology, University of Washington, Seattle, WA 98105, USA; 3Department of Environmental and Occupational Health Sciences, University of Washington, Seattle, WA 98105, USA

**Keywords:** inedible food waste, diet quality, HEI-2015, NRF, education, residential property values, food spending, ultra-processed foods

## Abstract

Americans waste about a pound of food per day. Some of this is represented by inedible food waste at the household level. Our objective was to estimate inedible food waste in relation to diet quality and participant socio-economic status (SES). Seattle Obesity Study III participants (*n* = 747) completed the Fred Hutch Food Frequency Questionnaire (FFQ) and socio-demographic and food expenditure surveys. Education and geo-coded tax-parcel residential property values were measures of SES. Inedible food waste was calculated from diet records. Retail prices of FFQ component foods (*n* = 378) were used to estimate individual-level diet costs. The NOVA classification was used to identify ultra-processed foods. Multivariable linear regressions tested associations between inedible food waste, SES, food spending, Nutrient Rich Food (NRF_9.3_) and Healthy Eating Index (HEI-2015) scores. Inedible food waste was estimated at 78.7 g/d, mostly from unprocessed vegetables (32.8 g), fruit (30.5 g) and meat, poultry, and fish (15.4 g). Greater inedible food waste was associated with higher HEI-2015 and NRF_9.3_ scores, higher food expenditures and lower percent energy from ultra-processed foods. In multivariable models, more inedible food waste was associated with higher food expenditures, education and residential property values. Higher consumption of unprocessed foods were associated with more inedible food waste and higher diet costs. Geo-located estimates of inedible food waste can provide a proxy index of neighborhood diet quality.

## 1. Introduction

Americans waste an estimated 422 g of food each day [1], including substantial amounts of healthy unprocessed meat and fish, dairy, vegetables, and fruit [2]. Wasted food contains wasted nutrients that could remedy the observed shortfalls in dietary intakes of under-consumed nutrients including fiber, calcium, and vitamin D [3,4]. There is also an economic component to household food waste. Based on the National Health and Nutrition Examination Survey (NHANES) data linked to national food prices, US consumers spend an average of $13.27 ($ United States dollars) per day on food [2]. Of the mean daily food expenditures, 59% represents the cost of consumed foods, 27% edible but wasted food, and 14% represents inedible food waste.

There are some key distinctions to be made between food loss, edible food waste, and inedible food waste, sometimes called food scraps [5,6,7]. Food loss occurs throughout the supply chain and can take place during production, postharvest handling, and processing [8]. Food waste typically occurs at later stages of the food supply chain, taking place at the retail and consumer household level [8]. Food waste can be avoidable or unavoidable. Unavoidable food waste at the household level is represented by inedible food scraps that can be composted but are typically hauled away by sanitation departments [9]. Inedible food waste has not been the topic of research studies. Rather, some key data have been generated by local sanitation departments. A recent audit of household disposals by the King County Solid Waste (KCSW) division shows that each King County household generates an estimated 49 pounds of compostable paper and food waste per month. The KCSW Organics Service has estimated that about 30.2% of that amount, equivalent to 14.82 lbs per month, came from inedible food waste, mostly from fresh vegetables and fruit [9].

Whereas most studies on food waste focus on edible food waste as a target for waste reduction efforts, few studies explore inedible food waste. This study used rich data from the Seattle Obesity Study III to estimate inedible food waste at the individual level in relation to multiple measures of diet quality and estimated diet cost. Food yields from the United States Department of Agriculture (USDA) Handbook 102 [10] were applied to 378 component foods of the Fred Hutch food frequency questionnaire used in the Seattle Obesity Study (SOS III). The present hypothesis was that inedible food waste, estimated from dietary records, would be an additional marker of diet quality. Diets containing more unprocessed foods, including fresh vegetables and fruit, would be ranked higher by the HEI-2015 score and would also generate more inedible food waste. Such diets would also be associated with higher reported food spending and higher estimated diet costs [11,12]. A secondary hypothesis was that higher levels of inedible food waste would be associated with higher socioeconomic status (SES). The SOS III participants were geo-localized to permit the mapping of diet quality and inedible food waste by geographic area.

## 2. Methods

### Study Design and Participants

The SOS III was a population-based longitudinal study of 872 adult residents of King, Pierce, and Yakima Counties in WA State [13,14]. Participant recruitment was county specific, relying on address-based sampling schemes stratified by property values along with community outreach to ensure broad representation by socio-economic status and race/ethnicity. Participant recruitment and data collection were conducted in-person by local staff at each research site from July 2016–May 2017.

Eligible adults aged 21–59 y, who were the principal food shoppers for the household, not pregnant or breastfeeding, and without any mobility issues, were invited to participate in the first in-person interview. Written consent was obtained before starting the study procedures. Data collection occurred during the first in-person visit, which was conducted at a local study site or at home (Yakima County only). Survey data were collected in English (in all 3 counties) and in Spanish (in Yakima County). All study procedures were approved by the Institutional Review Boards (IRBs) of respective study sites. The present analytical sample was based on 747 male and female respondents for whom complete dietary, socioeconomic, home address, and residential property values were available.

## 3. Methods and Procedures

### 3.1. Computer-Assisted Health Behavior Survey

A computer-based health and behavioral survey was used to collect data on socio-demographic variables, including age, sex, race/ethnicity, household income, education, employment status, marital status, and household size. Estimated monthly household food expenditures at home and away from home were obtained by self-report. At-home food expenditures referred to grocery purchases, whereas away-from-home food expenditures referred to foods consumed outside the home (e.g., restaurants, cafeterias). The estimated at-home and away-from-home food expenditures were summed to create a total monthly food expenditures variable. This was divided by household size to create monthly total food expenditures per capita.

Residential property values at the tax parcel level provided an additional measure of socioeconomic status [15]. Data from county tax assessors at the tax parcel level for 2016 were obtained for the three counties following previously published procedures [15].

### 3.2. Dietary Intakes Data and Inedible Food Waste

Dietary intake data were collected using the Fred Hutch Food Frequency Questionnaire (FFQ), also administered during the in-person interview. The FFQ consists of a list of 126 line-item foods, each of which is represented by a variable number of component food items that are weighted and used to estimate energy and nutrient content. The 126 goods are visible to the respondent but the underlying 384 component foods are not.

Inedible food waste was computed for 384 FFQ component food items using food yields based on the USDA Handbook 102 [10]. The USDA Handbook 102 provides yields following liquid loss and evaporation during cooking, but also estimates the amounts of peels, seeds, scraps, trimming, and stalks, which represent inedible food waste. The amount of inedible waste was standardized per 100 g of food.

The amounts of inedible food waste were then summed for each SOS III participant based on FFQ dietary intakes.

### 3.3. Energy-Adjusted Diet Cost

Estimates of individual-level daily diet cost were obtained by joining dietary intake data with county specific retail prices for 384 FFQ component foods. Retail prices were obtained from large supermarkets in King, Pierce and Yakima counties following standard and published procedures [11,16]. Retail prices converted to dollars per 100 g edible portion were added to Fred Hutch FFQ nutrient composition database, to parallel nutrient values, also expressed as amounts (g/mg/IU) per 100 g edible portion. The procedures of estimating diet costs from FFQ have been described previously [17]. For analytical purpose, this diet cost was divided by calorie intake and expressed per 2000 kcal/d. Diet cost per day was then converted into a monthly diet cost variable.

### 3.4. Percent Energy from Ultra-Processed Foods

The 384 FFQ component food items were aggregated into 4 NOVA food processing categories: unprocessed, processed, ultra-processed, and culinary ingredients, using published classification schemes [18]. Unprocessed foods have been defined as those fresh, dry, or frozen foods that had been subjected to minimal or no processing. The FFQ component foods included fresh meat, fish, fruits (such as apple, banana, apricots), salad, milk, vegetables (broccoli, green beans, potato), eggs, legumes, and unsalted nuts (raisins and prunes) and seeds. Culinary ingredients were sugar, animal fats (butter) and oils (olive oil, canola oil, corn oil), and salt [18]. Adding culinary ingredients (fat, sugar, salt) to wholesome fresh foods transforms them into processed foods. The FFQ component foods classified as processed foods included cheese, ham, beer and wine. FFQ foods classified as ultra-processed following NOVA criteria included breads, jams and jelly, breakfast cereals, sweet snacks (cookies and cakes), pizza, potato chips or tortilla chips, soft drinks (sodas and fruit drinks), French fries, sauces (ketchup, mayonnaise), desserts (ice cream, frozen yogurt, sherbet) and frozen meals, juices and soups.

Each of the 384 foods was coded as unprocessed, processed, ultra-processed or culinary ingredient, as documented previously [11,12]. The contribution of ultra-processed foods to total daily energy intakes was then calculated for each participant. This was done by dividing total energy intake from ultra-processed foods by total energy intake.

### 3.5. Diet Quality Measures

Diet quality was assessed by computing the Healthy Eating Index 2015 (HEI-2015), a measure of compliance with the 2015 Dietary Guidelines for Americans [19,20]. The HEI-2015 scores is derived using FFQ data, and is based on intake of 9 food groups to encourage (total fruit, whole fruit, total vegetables, greens and beans, whole grains, dairy, total protein, seafood and plant proteins and fatty acids) and 4 food groups to limit (refined grains, sodium and saturated fat and added sugars). The HEI-2015 is a continuous score on a scale of 0–100 where higher scores reflect higher diet quality. The score is adjusted per 2000 kcal.

### 3.6. Data Visualization

We generated high resolution choropleth maps of inedible food waste by census block for King County. To do this, we regressed inedible food waste in g/2000 kcal by tertiles of residential property values, adjusted for sex, age, race/ethnicity, and educational attainment. We did this based on the a priori assumption that food waste would display an SES gradient across residential priority tertiles, our proxy measure of SES, specifically accumulated wealth, similar to diet quality [15,21,22]. We then estimated the mean inedible waste in g/2000 kcal for each tertile of residential property value using predicted marginal using the sample mean value or mean proportion of covariates for King County respondents. Residential property values at the tax parcel level for all residential units in King County then aggregated by census block and were split into tertiles. Each census block was assigned a marginal mean inedible waste in g/2000 kcal for that tertile of property value from SOS III given that tertiles of property values for the SOS III King County sample were similar to that of the greater King County population. All GIS mapping and visualizations were conducted using ArcGIS Desktop release 10 [23].

### 3.7. Statistical Analysis

The present analysis used baseline dietary intake data from the SOS III study. Responses with missing data on sociodemographic variables, under and over-reporters of total energy intakes (<500 or >5000 kcal), and extreme outliers on estimated food expenditures were excluded. The final analytic sample size was 747 individuals.

Analyses of inedible food waste for each participant, expressed as g/day and g/2000 kcal, were conducted for the total sample and by socio-demographic groups of interest. Inedible food waste was also calculated by food group and by levels of diet cost. HEI-2015 values and percent energy from ultra-processed foods were calculated for each socio-demographic group. The association between inedible food waste and socio-demographics was tested using multiple-adjusted linear regression models with robust standard errors. Absolute inedible food waste was the dependent variable and sex, age, race, education, property value, and food spending were the independent variables. In multivariate model 1, food spending was represented by energy-adjusted diet cost; whereas in multivariate model 2 food spending was represented by self-reported food expenditures. Analyses was conducted using SPSS 22 (Armonk, NY, USA, v22.0) [24].

## 4. Results

Table 1 shows that the SOS III study sample was mostly female (81.9%), married (58.5%), evenly distributed by age group, and with 41% Hispanic participants, whereas 44.6% of the sample were college graduates, and 33.6% did not complete high school.

Mean inedible food waste (g/2000 kcal) was 85.5 g. There were significant differences by socio-demographic strata. Higher amounts of inedible food waste were associated with being female (*p* < 0.003), non-Hispanic Whites (*p* < 0.0001), higher education (*p* < 0.0001) and higher residential property values (*p* < 0.0001). There were no significant effects of age.

For each population subgroup, mean amounts of inedible food waste were directly associated with higher HEI-2015 diet quality scores and were inversely associated with percent energy from ultra-processed foods. In bivariate analyses, higher HEI-2015 score and lower percent energy from ultra-processed foods were associated with older adults, non-Hispanic Whites, college education, and higher residential property values. There were no significant effects of sex or marital status.

Table 2 shows the distribution of inedible food waste (g/day) by food groups and by food processing categories. Inedible food waste was estimated at 78.7 g/day. Among food group categories, inedible food waste mostly came from vegetables (32.8 g/day), fruit (30.5 g/day), and meat, poultry and fish (15.4 g/day). Among food processing categories, greater food waste was associated with diets with more unprocessed meat, poultry and fish and vegetables and fruit.

Figure 1 shows the relation between quintiles of residential property value and inedible food waste expressed as g/2000 kcal. The bottom quintile (Q1) of property value was associated with 72.4 g of inedible food waste; whereas, the top quintile (Q5) was associated with 94.5 g of inedible food waste.

Figure 2 shows the modeled map for tertiles of inedible waste (g/2000 kcal) for select areas of western King County. Mapping of inedible food waste geographically shows that neighborhoods with higher median residential property values such as waterfront areas along the Puget Sound, Lake Washington, and Mercer Island had higher inedible food waste. By contrast, areas with lower residential property values such as around South Seattle and near Seattle-Tacoma International airport had lower inedible food waste.

Table 3 shows inedible food waste and HEI-2015 by each food cost and diet quality indicator. Inedible food waste showed a dose response relation with cost. Going from the bottom to the top tertile of at-home cost expenditure, inedible food waste (g/2000 kcal) increased from 74.5 g to 98.6 g. Similarly, for diet cost, inedible food waste increased from 59.2 g to 118.8 g on going from the bottom to the top tertile. The mean HEI score increased by 9 points on going from the lowest to the highest tertile of diet cost (62.6 vs. 71.5). Consistent with these observations, inedible food waste increased significantly from the lowest tertile of HEI (61.2 g) to the highest tertile of HEI (107.5 g).

The results of multiple regression analysis between age, sex, race/ethnicity, education, residential property values, two measures of food spending and inedible food waste are shown in Table 4. In Model 1, higher diet cost was also associated with additional 55.9 g inedible food waste (*β* = 55.9, 95% CI = 48.26, 63.62). Having college education or higher was associated with 19.2 g more inedible food waste (*β* = 19.18, 95% CI = 8.68, 29.67) as compared to high school or less. Model 2, adjusted for self-reported food expenditures (in addition to SES) did not show much association with inedible food waste except with higher education (*β* = 20.7, 95% CI = 8.90, 32.42). Being female was also associated with more inedible food waste in both the models (model 1: *β* = 12.6, 95% CI = 4.82, 20.40; model 2: *β* = 18.84, 95% CI = 9.87, 27.82).

## 5. Discussion

The present analyses complement past studies by providing an estimate of inedible food waste at the consumer level. Based on analyses of NHANES 24-h recalls, Conrad et al. [1] estimated total edible and inedible food waste at 422 g/d. The present estimate of 78.7 g/day of inedible food waste was based on FFQs completed by SOS III participants. A recent study conducted in Canada reported 52 g inedible food waste per person daily [25].

The methods used by Conrad et al. [1] had some parallels with the present study. Both studies started with dietary intake data (24 h recall and FFQ) and both relied on USDA sources to estimate food waste. In the Conrad et al. [1] study, each food consumed by NHANES participants was disaggregated into its commodity ingredients by weight using the Food Commodity Intake Database [26], which were then linked to the USDA Loss-adjusted Food Availability data [27] to estimate edible and inedible food waste.

The present study relied on dietary records and the 384 component foods of the Fred Hutch FFQ. Each food item was adjusted for grams of food loss during preparation using values from the USDA Handbook 102, which lists yield factors for unprocessed foods, mainly vegetables, fruit and meat, poultry and fish. Whereas Conrad et al. [1] estimated the quantities of per capita food waste using loss-adjusted food availability (LAFA) data to calculate % of edible and edible waste for each underlying commodity, the present study used the USDA Handbook 102 to estimate yield factors for each individual food. One limitation of the USDA Handbook 102 is that it only allows the calculation of inedible food waste during food preparation at home.

Based on 2010 LAFA data, 10% of the total food supply is lost at the retail level and 21% is lost at the consumer level, excluding waste during production and processing. This was equivalent to 133 billion pounds in 2010. Assuming the 2010 US population to be 309.3 million, that would translate to a total of 535 g/person/day. This estimate of 535 g/d is higher than the Conrad estimate, as might be expected given differences between food supply and food consumption data.

Given mean household size of 3.4 people in the SOS III sample, we can estimate inedible food waste per household at around 4.48 lbs per week or 17.9 lbs per month for a household. Our results, from FFQ data adjusted using the USDA Handbook 102 on food yields, show good agreement with the King County Solid Waste (KCSW) Organics division sanitation audit, which estimated that 14.8 of the average 49 lb of food waste and compostable paper per household per month consisted of inedible fruit vegetables and meat [9].

The same connection was made by Carroll et al. [25] who used a combination of dietary intake records and household audits of food waste. Diet quality was assessed using the Healthy Eating Index-2015 (HEI-2015), calculated from 3-day food records. Household food waste per capita was calculated based on detailed waste audits conducted over multiple weeks. The study showed strong associations between parent and child HEI-2015 scores and daily per capita edible (avoidable) and inedible (unavoidable) food waste, adjusting for income. The combined use of 3-day food records and household audits was novel and the results are supported by the present data.

Estimating food expenditures at home has acquired a new importance, given continuing limitations on away-from-home eating due to the ongoing COVID-19 pandemic [28,29] and the current rise in food preparation at home [29]. Our results linking SES with inedible food waste support prior findings by Conrad et al. [1] that higher quality diets in the NHANES sample were associated with greater total food waste among American adults. In the present study, significant positive associations were observed between two diet quality metrics and inedible food waste from unprocessed meat and fresh vegetables and fruit. Diets that were higher in unprocessed foods received higher HEI-2015 and NRF_9.3_ diet quality scores but also tended to be more expensive [11]. The socioeconomic gradient in diet quality meant that higher levels of inedible food waste were found among higher SES groups living in more affluent neighborhoods.

To our knowledge, this is the first study to explore the use of inedible food waste as a potential index of diet quality, with direct links to food spending and household SES. In the present study, we used two measures of food spending: food expenditures obtained through self-report and estimated individual level diet costs. Both indicators showed positive association with food waste in bivariate analyses; however, after adjusting for covariates, only calculated estimates of diet cost (rather than self-reports) showed positive association with food waste. There was also a significant association between inedible food waste and geo-coded residential property values. Such geocoding opens the door to mapping the generation of compostable food waste across neighborhoods. Though this study was conducted in Seattle, the methods are transferable to other geographic areas.

Inedible food waste at the household level as assessed by local waste collection agencies could be a potential indicator of neighborhood diet quality. Obtaining empirical estimates of community-level inedible waste would be challenging: not all municipalities collect compostable goods separately from other waste streams; disaggregation of household waste by community may not be feasible depending on when, where, and how household waste is combined; and determining the proportion of inedible versus edible food waste may not be part of routine data collection for waste management agencies. However, occasional sanitation audits that are designed with these considerations in mind could open up the possibility of using empirical estimates of community-level inedible food waste from waste management agencies as proxy indicators for community-level diet quality.

The potential use of inedible food waste as a proxy indicator for neighborhood diet quality could aid in describing geographic patterns in diet quality or serve as a downstream indicator of the effectiveness of interventions that promote the preparation of unprocessed foods. The Waste and Resources Action Programme (WRAP), which led household food waste reduction campaigns in the United Kingdom, demonstrated the effectiveness of their interventions by showing decreases in household food waste during the intervention period, as measured by curbside waste collection from local authorities [30]. In the case of WRAP, decreased total food waste as measured by waste collection agencies reflected the success of a food waste reduction intervention. Could an increased proportion of inedible food waste (compared to edible food waste) in waste collection streams indicate the success of an intervention to improve diet quality?

Inedible food waste is by its nature, inevitable; and by the nature of its linkage to diet quality, it may be welcomed. Widespread improvements in diet quality might necessitate strategies to prepare municipal infrastructure to handle an increased volume of inedible food waste. WRAP’s food waste reduction campaigns were accompanied by efforts to increase the availability of separate household food waste collections in the United Kingdom [31]. In the European Union, the Waste Framework Directive will make separate food waste collection obligatory by the year 2023 [32].

A survey of residential waste collection programs in the US identified only 148 communities with curbside food waste pickup and 67 communities with food waste drop-off programs as of 2017 [33]. As of 2017, 6.3% of food waste generated in the United States was composted [34], with the remaining food in the municipal solid waste stream where it was incinerated or landfilled. This is problematic given that organic matter emits methane during anaerobic decomposition in landfills; with one third of all food lost or wasted globally [35], food waste accounts for 6% of global greenhouse gas emissions [32,36].

Globally, reductions in food waste and more sustainable waste management practices will contribute to the United Nations Sustainable Development Goal 12, “responsible consumption and production” [37]. The Sustainable Development Goals also prioritize diet quality. Taken together, these issues underscore the importance of working across sectors to address societal goals with unexpected interdepencies. In this case, improving diet quality while minimizing the environmental impact of food waste—including inedible food waste—may require unexpected partnerships between the public health and waste management sectors.

The present study adds to the continuing dialogue on topics that are vital to food security and at the national level [1,2,3]. Spiker et al. [3] calculated the nutritional value of retail- and consumer-level waste for 213 commodities in the US Department of Agriculture Loss-Adjusted Food Availability data series. The estimated losses for 2012 were 1217 kcal, 33 g protein, 5.9 g dietary fiber, 1.7 μg vitamin D, 286 mg calcium, and 880 mg potassium per capita per day; these losses represent non-trivial proportions of recommended intakes, including for nutrients that are under-consumed in the United States.

The present study had limitations. First, the present sample was limited to countries in WA State and was not nationally representative. Second, dietary intakes were based on FFQ data, rather than multiple 24-h recalls. However, FFQs have been widely used in nutritional epidemiological studies, including SOS. Third, diet cost estimates did not represent actual food expenditures; rather they were obtained by merging dietary intakes data with retail food prices, a method also used by Conrad et al. [2]. They represent the average supermarket prices at the time, in conjunction with component FFQ foods. To address this limitation, we used participants’ self-reported food expenditures as an additional indicator. Fourth, ambiguity in the NOVA classification system may have resulted in some misclassification, though this has been minimized by employing two independent researchers to assign food items to a NOVA category. Additionally, the present study was based on cross-sectional data; hence, associations observed between SES, diet cost and other diet quality indicators cannot be causally interpreted.

The study had several strengths. The study design, by looking at inedible food waste separately from edible food waste, enables a more nuanced exploration of household waste streams that goes beyond total food waste. The amount of inedible waste per household was estimated using dietary intake and food yield data and was comparable to empirical estimates from a KCSW sanitation audit. The modelled heat maps of compostable food waste by the Seattle neighborhood represent a novel means of viewing geographic distribution of inedible compostable food waste across neighborhoods.

## 6. Conclusions

In this study, more inedible foods waste was associated with higher diet quality; higher food expenditures, education, and residential property values; and higher consumption of unprocessed foods. The use of geo-located data facilitated the mapping of inedible food waste across neighborhoods. The link between inedible food waste and diet quality suggests that empirical measures of inedible food waste, if obtained through local waste collection agencies, could potentially serve as a proxy indicator for neighborhood-level diet quality. Methods for doing so would need to be further studied and validated, but this work illustrates the potential for synergy between multi-sectoral efforts to improve human health outcomes and implement sustainable waste management. 

## Figures and Tables

**Figure 1 nutrients-13-00479-f001:**
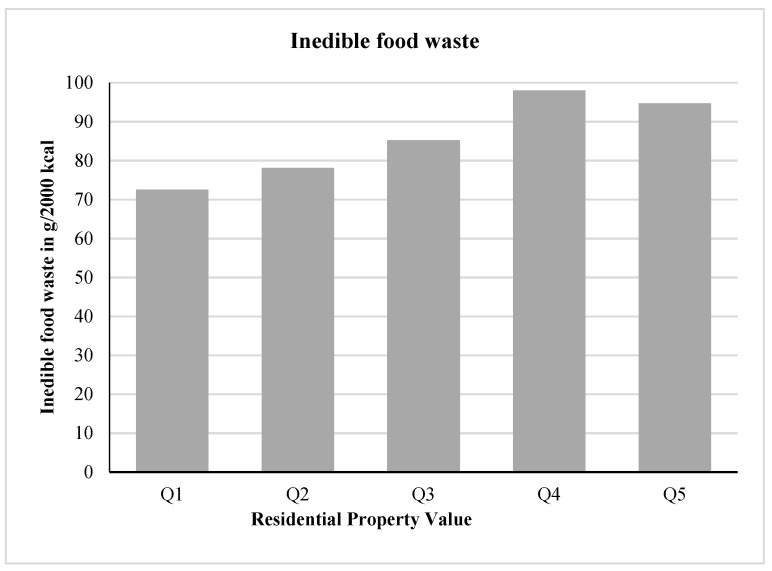
Inedible food waste (g/2000 kcal) and quintiles of residential property value.

**Figure 2 nutrients-13-00479-f002:**
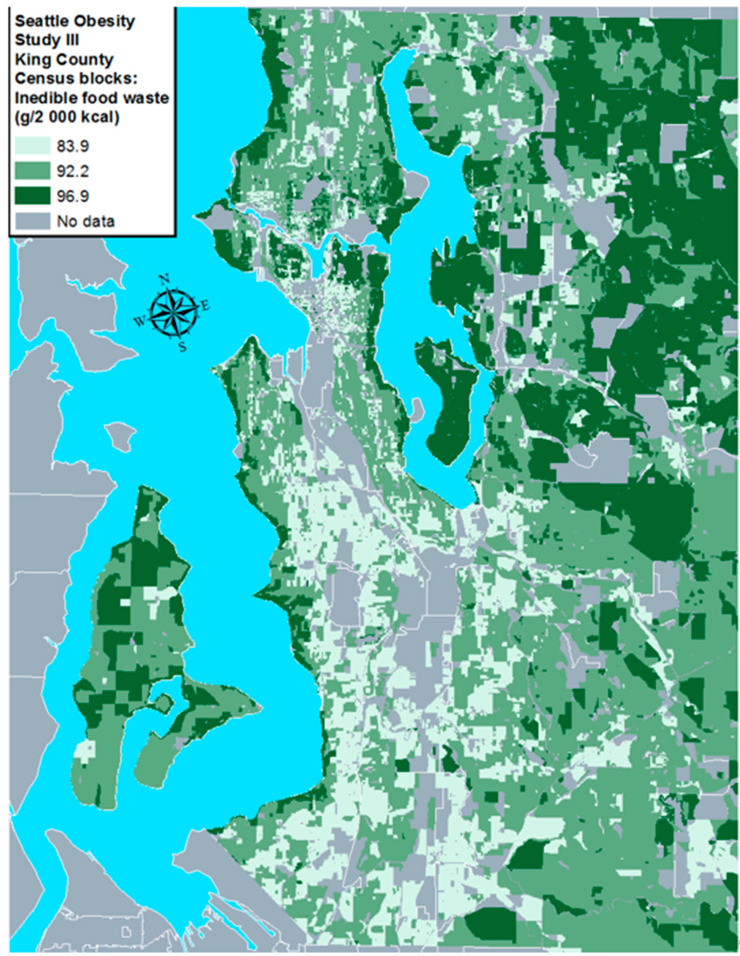
Mean inedible food waste by census block based on multivariate models regressing inedible food waste (g/2000 kcal) by tertiles of residential property values at the individual level, controlling for age, gender, race/ethnicity, and educational attainment, King County, Washington.

**Table 1 nutrients-13-00479-t001:** Inedible food waste (g/200kcal/per capita) by socio-demographic variables.

Variables	Total		Inedible Food Waste (Grams per Day)			Inedible Food Waste (Grams per 2000 kcal)			HEI 2015Diet Quality Score			% Energy from Ultra-Processed Foods		
	*n*	%	Mean	SD	*p*-Value	Mean	SD	*p*-Value	Mean	SD	*p*-Value	Mean	SD	*p*-Value
Overall	747	100	78.7	42.73		85.5	48.78		67.1	9.91		59.6	10.75	
Sex														
Male	135	18.1	75.0	43.99	Ref.	74.2	44.22	Ref.	66.5	10.33	Ref.	58.4	10.46	Ref.
Female	612	81.9	79.6	42.66	0.255	88	49.41	0.003	67.2	9.82	0.48	59.9	10.8	0.13
Age														
21–40	283	37.9	79.1	44.48	Ref.	85.4	53.4	Ref.	66.1	10.13	Ref.	60.2	11.08	Ref.
41–50	229	30.7	76.5	40.10	0.489	82.2	41.9	0.442	66.5	9.75	0.683	60.5	9.9	0.719
>51	235	31.5	80.4	43.17	0.734	88.9	49.2	0.4	68.8	9.62	0.002	58.2	11.02	0.039
Race/ethnicity														
Non-Hispanic White	362	48.50	83.7	45.89	Ref.	91.80	52.90	Ref.	69.6	10.15	Ref.	57.6	11.04	Ref.
Hispanic	303	40.60	72.8	35.77	0.001	76.20	37.62	0.0001	64.3	8.93	0.0001	63.3	8.69	0.0001
Other	82	11.00	78.7	49.22	0.399	92.40	59.75	0.931	66.1	9.37	0.002	55.1	12.29	0.081
Marital Status														
Married	437	58.5	82.4	42.66	0.005	88.70	46.02	0.04	67.7	9.65	0.055	59.4	10.77	0.422
Single	310	41.5	73.5	42.36	Ref.	81.00	52.16	Ref.	66.2	10.23	Ref.	60	10.73	Ref.
Education														
High school or less	251	33.60	70.2	32.67	Ref.	72.30	33.77	Ref.	63.6	9.22	Ref.	64.1	8.44	Ref.
Some College	163	21.80	77.1	46.52	0.102	81.10	49.85	0.05	66.1	9.9	0.011	60.5	10.15	0.0001
College or more	333	44.60	86.0	46.20	0.0001	97.60	54.66	<0.0001	70.2	9.48	0.0001	55.9	11.24	0.0001
Residential Property value														
Tertile 1 (<$130k)	249	33.30	71.6	33.73	Ref.	74.00	35.04	Ref.	64.3	9.1	Ref.	63.9	8.3	Ref.
Tertile 2 ($130–293k)	249	33.30	77.6	46.31	0.095	87.20	55.36	0.0001	67.2	10.16	0.001	59.1	11.13	0.0001
Tertile 3 (>$293k)	249	33.30	87.1	45.69	0.0001	95.30	51.31	0.0001	69.8	9.69	0.0001	55.9	11.07	0.0001

HEI = Healthy Eating Index, *n* = sample size, SD = standard deviation, Ref. = reference group, $ United States dollars.

**Table 2 nutrients-13-00479-t002:** Daily per capita inedible food waste (g/day) across food groups and food processing categories.

Food Categories	Mean (g/day)	95% CI
Total daily food waste	78.7	(75.6–81.8)
Food Groups		
Dairy and eggs	0	
Meat, poultry and fish	15.4	(14.7–16.2)
Beans, nuts, and seeds	0	
Grains, cereals	0	
Vegetables	32.8	(31.1–34.4)
Fruits	30.5	(28.5–32.4)
Fats and sweets	0	
Food processing categories		
Unprocessed	76.0	(72.9–79.1)
Processed	0	
Ultra-Processed	2.6	(2.4–2.8)
Culinary Ingredients	0	

CI = confidence interval.

**Table 3 nutrients-13-00479-t003:** Inedible food waste by indicators of food spending (in US $/month) and diet quality.

Variables	Total		Inedible Food Waste (Grams per 2000 kcal)			HEI 2015		
	*n*	%	Mean	SD	*p*-Value	Mean	SD	*p*-Value
Overall	747	100	85.5	48.78		67.1	9.91	
**Food spending** **at home ($/month)**								
≤$100	273	36.5	74.5	40.57	Ref.	65.2	9.96	Ref.
>$100–≤175	231	30.9	84.7	45.18	0.008	66.4	9.88	0.177
≥$175	243	32.5	98.6	56.86	0.0001	69.9	9.27	0.0001
**FFQ diet cost ($/month)**								
≤$252	249	33.3	59.2	24.28	Ref.	62.6	10	Ref.
>$253–≤299	249	33.3	78.5	34.28	0.0001	67.1	9.77	0.0001
≥$300	249	33.3	118.8	59.47	0.0001	71.5	7.81	0.0001
**HEI-2015 score**								
Tertile 1	249	33.3	61.2	32.72	Ref.			
Tertile 2	249	33.3	87.9	48.78	0.0001			
Tertile 3	249	33.3	107.5	51.19	0.0001			

$ United States dollars, HEI = Healthy Eating Index, *n* = sample size, SD = standard deviation, Ref. = reference group.

**Table 4 nutrients-13-00479-t004:** Linear regression analyses of inedible food waste by sociodemographic indicators.

Variables	Model 1			Model 2		
	Coeff	*p*-Value	95% CI	Coeff	*p*-Value	95% CI
Sex						
Male	Ref.			Ref.		
Female	12.61	0.002	(4.82, 20.40)	18.84	0.0001	(9.87, 27.82)
Age						
21–40	Ref.			Ref.		
40–50	−6.73	0.063	(−13.84, 0.38)	−4.86	0.240	(−12.97, 3.24)
>50	−0.80	0.844	(−8.78, 7.18)	−2.67	0.570	(−11.87, 6.54)
Race/ethnicity						
NonHispanic White	Ref.			Ref.		
Hispanic	−1.12	0.874	(−14.88, 12.65)	5.76	0.468	(−9.80, 21.31)
Other	3.02	0.604	(−8.39, 14.43)	4.39	0.506	(−8.55, 17.33)
Education						
High school or less	Ref.			Ref.		
Some College	3.04	0.513	(−6.07, 12.14)	4.89	0.355	(−5.46, 15.24)
College graduate/Grad school	19.18	0.0001	(8.68, 29.67)	20.66	0.001	(8.90, 32.42)
Residential Property value						
Tertile 1 (<$128,675)						
Tertile 2 ($128,675.20–$290,866)	1.22	0.774	(−7.12, 9.56)	5.44	0.287	(−4.57, 15.45)
Tertile 3 (>$290,866)	0.46	0.933	(−10.11, 11.02)	6.22	0.314	(−5.90, 18.33)
Food expenditure ($/month)						
≤$142.6	Ref.			Ref.		
>$142.6–≤250				−4.30	0.256	(−11.73, 3.13)
≥$250				11.22	0.042	(0.42, 22.03)
FFQ diet cost ($/month)						
≤$320	Ref.			Ref.		
>$320–≤400	18.44	0.0001	(12.58, 24.30)			
≥$400	55.94	0.0001	(48.26, 63.62)			

$ United States dollars, HEI = Healthy Eating Index, SD = standard deviation, Ref. = reference group.

## Data Availability

Data presented in this study are available on request from the corresponding author. Geo-localized data are not publicly available due to the need to maintain participant confidentiality.

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
