# Peer review of "Inedible Food Waste Linked to Diet Quality and Food Spending in the Seattle Obesity Study SOS III"

_nutrients, 2021, doi:10.3390/nu13020479_

Round 1

Reviewer 1 Report

The authors undertake a rigorous process and arrive at compelling findings. Further analysis would be helpful of implications and alternatives to compostable food waste and not composting, but sending waste to landfill.

Reviewer 2 Report

The authors estimate inedible food waste at the individual level in relation to measures of diet quality and estimated diet cost. This is a very interesting, novel, and relevant topic. This is a clear and well written article. I have some suggestions for your manuscript.

Introduction:

The rationale for the study was well explain. Your introduction could better inform the reader why the paper is important in terms of the relationship between food waste, diet quality, health inequities and environmental sustainability. The authors should demonstrate what new knowledge is brought by their study (and originality). How can food waste management achieve sustainable development goals?

Methods:

Why were not height and weight measurements included in the analyses? And was food insecurity assessed? If not, please indicate as a limitation of the study. If yes, please include in the results.

Results:

How many eligible individuals refused to participate? Please indicate participation rate. Please provide information considering the comparison between participants and refusals and excluded individual.

Discussion:

Please discuss the use of individual and household measures.

The discussion seems to be descriptive. Fresh insights about the role of waste management in the reduction of food and health inequities are missing.

Implications for waste management policies and best practices in food waste prevention should be highlighted.

Conclusion:

It seems that a concluding sentence to highlight the importance of mapping compostable food waste to the global and multidimensional problem of food waste in ethical, social,  environmental, and economic terms, is missing.

Reviewer 3 Report

I have read this manuscript with great interest. However, after reading I have some doubts. I have presented them first as general and then as specific comments.

Introduction

Line 34 -56. This chapter describes the phenomenon of food waste in the USA, and explains the concepts related to food waste. Information about the own study appears next. There is a lack of information about the results of previous studies conducted in other populations, why knowledge on this subject may be important, and what this study contribute to science.

I wonder about the usefulness of the assessment of inedible food waste, estimated from dietary records, as an additional marker of diet quality. With the information from dietary records, it is easy to get to know diet quality. Please explain it more.

The hypothesis that diets containing more unprocessed foods, including fresh vegetables and fruit, would be ranked higher by the HEI-2015 and the NRF9.3 diet quality scores seems fairly obvious. It is worth confirming this in the introduction by recalling the results of research in this field. Also, such a diet must generate more inedible food waste due to the presence of fresh vegetables and fruits. I suggest to rethink this hypothesis.

I wonder if it is important for this article that the SOS III participants were geo-localized to permit the mapping of diet quality and inedible food waste by geographic area. This makes the results even more local. Such mapping seems to be more important from the perspective of food waste management, but then the article should be prepared differently

Methods and procedures

Line 91-92 How the estimation was made, i.e. what was the question?

Line 126-127 I do not understand this sentence : “The 126 goods are visible to the respondent but the underlying 384 component foods are not”. Please explain it by giving examples.

Line 112 – 113 This information is not necessary

Lines 122 – 123 “In this way, each of the 384 foods in the G-SEL database was associated with 45 nutrient vectors and a single cost vector”. In my opinion we do not need this information. If yes, please explain the abbreviation G-SEL, also “45 nutrient vectors”. If this information is in reference 17, it is redundant for this manuscript.

Lines 147 – 154 I suggest to reformulate the text to indicate how the quality of the diet was assessed – the change in style.

In the abstract is written that education and geo-coded tax-parcel residential property values were measures of SES. However in this section I did not find information on it. And in the manuscript SES is not used anymore.

Results

Lines 191-199 – The change in style is needed because it is not a legend to the Table

Lines 194-195 – It is only description - these results should be presented in Table.

Lines 214 – 223 – I suggest to omit this part. It is important from the perspective of waste management, not the quality of the diet. See my comment in Introduction section.

Line 215 “Clear geospatial patterns of inedible waste emerged” – how were these patterns assessed?

Lines 233-240. Sex was also associated with inedible food waste (in both models) and food expenditure in model 2 (p = 0.04). It was not mentioned in the description of the results.

Discussion

Line 245. Information on the amount of inedible food waste is needed.

Lines 248- 255. In my opinion the comparison of both study method can be omitted without any loss for the manuscript.   

Lines 303-304. “Such geocoding opens the door to mapping the generation of compostable food waste across Seattle neighborhoods”. What about international perspective?

There is no reference to the world situation in the discussion. The focus is only on the US. Too much attention is focused on discussing the methods of obtaining the information, and little space is devoted to the results obtained, e.g. gender versus inedible waste, etc.

Conclusion

Only inedible food waste was assessed in the study. Thus, conclusion concerning total food waste is not justified.

Some detailed comments:

Line 5 without „and”

Line 23 the abbreviation NRF9.3 needs description

Line 91-92 How the estimation was made, i.e. what was the question (s)

Line 99 the references should be written similar to others

Line 100  I suggest to omit one of the “for  2016”

Line 137 – the tense of transform should be consider

Round 2

Reviewer 3 Report

Thank you very much for taking into account some my comments in new version of the manuscript. Nevertheless, I still feel that the introduction is not a good introduction to the results presented in the manuscript. There was no mention at all of the fact that no one, or almost no one, rated the share of inedible food waste and why it is so important.

Author Response

Thank you for the comment! We have now revised the introduction to address this suggestion.